# Government subsidies and innovation in new energy vehicle companies: An empirical study of new energy vehicle listed companies based on Shanghai and Shenzhen A-shares

Jianguo Sun[1], Mingfu Tian [1]*, Weitong Zhang[2], Jingyi Ning[3]

**1** School of Economics, Henan University, Kaifeng, Henan, China, **2** Institute of Zhongyuan Development, Henan University, Kaifeng, Henan, China, **3** School of Business, Zhengzhou University, Zhengzhou, Henan, China

* tianmingfu@henu.edu.cn

## Abstract

The panel data of 50 new energy vehicle enterprises in Shanghai and Shenzhen A-shares from 2012 to 2021 are selected to empirically analyze the impact of government subsidies on the innovation of new energy vehicle enterprises and to further discuss the differences between such an impact in different forms and regions. The study finds that, first, government subsidies have a certain promotion effect on the innovation of new energy vehicle enterprises, and an inverted U-shaped relationship exists thereof. Second, at the enterprise level, government subsidies have a significant effect on the innovation of non-state enterprises, downstream vehicle enterprises, and enterprises with lower establishment years, and the inverted-U trend is evident. Third, at the regional level, government subsidies have a more significant effect on the innovation of enterprises in non-eastern regions and low-environmental regulation regions, and the inverted-U-shaped relationship trend is more apparent. The study establishes the nonlinear relationship between government subsidies and the innovation of new energy vehicle enterprises through empirical research, which expands the theory of enterprise innovation and has a certain guiding significance for improving the innovation capability of new energy vehicle enterprises in the future.

## 1. Introduction

Owing to the unique external advantages of new energy vehicles in connection with energy conservation, emission reduction, and low-carbon environment protection [1, 2], they are gradually becoming an emerging industry, strongly supported by the world nowadays. Moreover, promoting the development of the new energy vehicle industry is a robust approach toward seizing the high ground of science and technology innovation and competing for the national environmental protection discourse [3, 4]. China's rapid GDP growth has been accompanied by increasingly severe environmental pollution problems [5], and the development and popularization of new energy vehicles are of great significance for China to cope

**Data Availability Statement:** All relevant data are within the paper and its Supporting information files.

**Funding:** The author(s) received no specific funding for this work.

**Competing interests:** The authors have declared that no competing interests exist.

with environmental problems and achieve the goal of carbon neutrality and carbon peaking. Unlike traditional industries, the new energy vehicle industry, as a high-tech industry, faces multiple challenges, such as high production costs, large investments in research and development (R&D), and large market uncertainty [6]. Therefore, government departments need to provide certain policy support through macro-control, especially in the early stages of development. China launched the "863" program in 2001 to help develop new energy vehicles, an innovation that ushered in the development of China's new energy vehicles, and by 2022, China's new energy vehicle production and sales ranked first worldwide. Although the overall volume of China's new energy vehicles is large, the development process has caused some controversy. The subsidy-based form of support inevitably induces some market failures, such as the "rent-seeking" behavior of enterprises due to subsidies, and special subsidies cannot be earmarked for specific purposes. Furthermore, the frontier technology innovation capability remains slightly inadequate, as indicated by the "lack of core" of new energy vehicles [7]. These problems seriously restrict the pace of upgrading China's new energy vehicle industry.

Determining whether government subsidies play a role in promoting new energy vehicle enterprises is an extensively discussed topic in the current research on new energy vehicles. The existing literature roughly divides the mechanism of the role of government subsidies on enterprise innovation into three broad categories. First, government subsidies have incentive effect on enterprise innovation [8]. Second, government subsidies inhibit enterprise innovation [9]. Third, there is no significant effect [10]. However, from a nonlinear perspective, this study explores the nonlinear relationship between government subsidies and new energy vehicle enterprises based on the linear relationship, which enriches the innovation theory of new energy vehicle enterprises, further expands the study of nonlinear mechanisms, and provides evidence for the inverted U-shaped relationship between government subsidies and enterprise innovation in China's new energy vehicle industry.

Different types of enterprises are affected differently by government subsidies. As the enterprises may receive different degrees of support from government departments due to the nature of property rights [11], industrial chain division [12], and enterprise age [13], the policy effects may differ. Herein, we divide the enterprises into three industrial chains—upstream, middle, and downstream—which are more detailed and thorough relative to the existing literature. In addition, the location of the enterprise is also an important factor that influences the effect of the policy [14]. Some studies show regional imbalances in government subsidies [15]. This can lead to various marginal effects due to differences in regional innovation factors and government resource tilting. In this study, when considering regional factors, the intensity of environmental regulations is considered for two reasons. On the one hand, environmental regulations may considerably reflect the degree of regional aspirations for developing new energy vehicles. On the other hand, the intensity of environmental regulations can reflect the implementation of government department policies from the side. Most existing literature only considers the geographical division of the region while ignoring the differences in policy implementation, which are considered in this study.

How do government subsidies affect innovation in new energy vehicle companies? How are different types and regions of firms affected? This study analyzes the mechanism of government subsidies on the innovation of new energy vehicle enterprises and the differences in the influence on different types of enterprises and regions from a quantitative perspective, based on panel data of enterprises from 2012 to 2021, which enriches the research on new energy vehicle micro-enterprises. This is especially of great reference value for the policy formulation and stimulation of enterprise innovation of new energy vehicles in China. The remainder of this paper is organized as follows: Section 2 presents the literature review and theoretical analysis, Section 3 focuses on the research design, Section 4 analyzes the empirical results, Section 5

presents the heterogeneity analysis, and, finally, Section 6 provides the conclusion and policy implications.

## 2. Literature review and theoretical analysis

### 2.1 Government grants and business innovation

Owing to the significant externalities associated with new energy vehicles, government innovation incentives in the form of government subsidies to enterprises help them achieve industrial upgrading, optimize resource allocation, and regulate market failures [16]. The development of China's new energy vehicle industry currently relies mainly on government subsidies and technological innovation [17], and research on new energy vehicle micro-enterprises is basically centered on these two aspects. In the early stages of development, new energy vehicle enterprises face the dual pressures of low-cost competition from traditional fossil energy and the high cost of new energy [18]. Government subsidies can relieve the pressure of capital investment and financing restrictions [19], in addition to signaling R&D institutions, promoting cooperation between enterprises and R&D institutions, and improving the level of enterprise innovation [20]. However, in the long run, it may lead to the "subsidy dependence" of enterprises, which gives rise to difficulties in breaking through core technologies and overcapacity [21].

Research on the incentive effect of government subsidies on enterprise innovation are abundant. They show that government subsidies can directly or indirectly compensate for the shortage of enterprise innovation investment, effectively reduce the cost and risk of enterprise innovation, and facilitate enterprise innovation [22]. However, there are also views that government subsidies have a "crowding out" effect on enterprise innovation, thereby inhibiting enterprise innovation behavior, and government subsidies have a restrictive effect on enterprise technological innovation [17]. Additionally, some scholars point out that due to information asymmetry, the government tends to select enterprises with low risk and high short-term returns when screening target enterprises, which makes those enterprises with low short-term returns but good long-term development benefits often fail to receive subsidies [23]. The purpose of government subsidies is essentially to reduce the cost and potential risks of enterprise innovation through financial support and compensate for the shortage of enterprise innovation factors in response to market failure. However, inappropriate and excessive subsidies may result in the government paying for the inputs that should be provided by market mechanisms for enterprise technological innovation, and selective subsidy policies may cause the "crowding out" effect. Therefore, government subsidies need to be kept within a certain range, which can effectively promote enterprise innovation when the subsidies are below the threshold [24] and be counterproductive when they exceed the threshold [25]. Moreover, excessive government subsidies may also increase the risk of enterprise "rent-seeking" and "subsidy fraud," as well as increase the inertia of enterprises and generate negative externalities. Therefore, there may be a nonlinear mechanism between government subsidies and new energy vehicle enterprises, and the following hypothesis is proposed:

H1: There is an inverted U-shaped relationship between government subsidies and the innovation output of new energy vehicle enterprises, which is first incentivized and then inhibited.

### 2.2 Enterprise level heterogeneity

Under China's special economic system, state owned enterprises (SOEs) and non-SOEs are subject to different institutional restrictions, project approval processes, and access to government subsidies [26]. Owing to the government's "underwriting," SOEs have certain advantages

over non-SOEs in terms of market competition. Even if the marginal output of SOEs' innovation is low, they will not be eliminated from the market. Non-SOEs, as self-sufficient market players, are subject to market laws and regulations in a perfectly competitive market, but they have to continuously innovate themselves to adapt to market competition. This self-financing model highlights the incentive effect of government subsidies on enterprise innovation and reduces the risk of failure and the cost of institutional regulation [27].

The new energy vehicle industry chain is a complex ecosystem wherein upstream, midstream and downstream enterprises interact with each other, and a clear industrial division of labor system helps to study the innovation mechanism of new energy vehicle enterprises more thoroughly [28]. According to the division of labor in the new energy vehicle industry chain, the industry chain can be roughly divided into upstream, raw material development and utilization; midstream, core components and R&D and manufacturing of batteries, motors, and electric controls; and downstream, vehicle manufacturing and sales. With downstream vehicle enterprises as an important hub in the new energy vehicle industry chain, the innovation results will directly affect consumers, and better innovation results will receive positive feedback from consumers, which will lead to greater marginal gains and continuous technology iteration, forming a virtuous cycle of innovation [29].

As an important indicator of the maturity of a firm's development, the number of years of firm growth has been a factor that many scholars should consider in influencing firm performance. The development of new energy vehicles in China is just over twenty years old. In general, emerging firms are becoming more active in the market; their market share is increasing; and their incentive to innovate is increasing, but they are less likely to receive government subsidies, probably due to their weaker political connections. Older enterprises have achieved economies of scale and may receive more government subsidies because of their strong political connections and monopoly position in the market, resulting in "subsidy fraud" and political rent-seeking behavior, which reduces the incentive for innovation [30] The following hypothesis is proposed:

H2: Government subsidies have a greater stimulating effect on the innovation of non-state owned, downstream industry chain, and less established new energy vehicle companies, with a more obvious trend of inverted "U" shaped relationship.

## 2.3 Regional level heterogeneity

On account of the higher level of economic development in the eastern region of China, enterprises generally have a higher level of innovation [31], and more diverse financing channels. The concentration of manufacturing industries in the eastern region also generates spatial agglomeration effects and positive externality effects, which lead to a higher level of innovation among enterprises in the eastern region and a higher threshold of policy stimulation, while the approval of subsidies in the eastern region is more stringent compared to the non-eastern region. As for the non-eastern regions, due to the inherent shortage of resource factors, government subsidies can quickly help their development and make up for the shortage of resources [32].

The intensity of environmental regulation reflects, to a certain extent, the level of governance and development in the region. Most of the regions with high environmental regulation belong to the more dense manufacturing regions, which have urgent requirements for the development of new energy vehicles and pay more attention to the promotion of innovation in new energy vehicles. This to a certain extent reduces the occurrence of market failure, while most of the regions with low environmental regulation belong to the less developed regions, where the development of new energy vehicles is at a preliminary stage and the government

subsidies are more rugged and the incentive effect of the policy is more obvious [33]. The following hypothesis is proposed:

H3: Government subsidies have a greater effect on stimulating innovation in non-eastern, low-environmental-regulation regions, with a more pronounced inverted "U"-shaped relationship.

## 3. Research design

### 3.1 Sample selection

Considering that the subsidies for new energy vehicle enterprises began to gradually decline after 2012, which may affect the innovation output of new energy vehicle enterprises, this study selects new energy vehicle enterprises in Shanghai and Shenzhen A-shares from 2012 to 2021 as the research sample. To accurately distinguish new energy vehicle enterprises from traditional fuel vehicle enterprises, we refer to the division of new energy vehicle enterprises by the Ministry of Industry and Information Technology in conjunction with the *YEARBOOK OF ENERGY-SAVING NEW ENERGY VEHICLE*. Fifty listed new energy vehicle enterprises are selected as the research sample, among which 10 belong to upstream enterprises, 16 belong to midstream enterprises, and 24 belong to downstream enterprises (Table 1).

### 3.2 Model setting

To further investigate the relationship between government subsidies and firm innovation, based on the relevant theoretical analysis, the following static panel estimates are constructed, where, M1 is estimated linearly and M2 is estimated nonlinearly by adding a quadratic term of government subsidies on the base of M1.

$$M1 : Patent_{it} = \beta_0 + \beta_1 Subsidy_{it} + \beta_2 X_{it} + \gamma_t + \mu_i + \sigma_{it}$$

$$M2 : Patent_{it} = \beta_0 + \beta_1 Subsidy_{it} + \beta_2 Subsidy_{it}^2 + \beta_3 X_{it} + \gamma_t + \mu_i + \sigma_{it}$$

Considering that the innovation output of the previous period may have an impact on the innovation output of the current period, the following dynamic estimated panel model is constructed with reference to Li et al. [34], where, M3 is a linear dynamic panel model and M4 is a nonlinear dynamic panel model.

$$M3 : Patent_{it} = \beta_0 + \beta_1 Patent_{it-1} + \beta_2 Subsidy_{it} + \beta_3 X_{it} + \gamma_t + \mu_i + \sigma_{it}$$

$$M4 : Patent_{it} = \beta_0 + \beta_1 Patent_{it-1} + \beta_2 Subsidy_{it} + \beta_3 Subsidy_{it}^2 + \beta_4 X_{it} + \gamma_t + \mu_i + \sigma_{it}$$

Patent is the dependent variable, which indicates the innovation output of the new energy vehicle enterprises in the current year. Subsidy is the core explanatory variable, which indicates the government subsidies received by the enterprises in the current year. X is a series of control variables, including R&D expenditure (Expenditure); R&D staff input (Staff); the year of establishment (Age); the asset and liability ratio (Debt); Tobin Q (Tobin Q); the shareholding ratio of the first largest shareholder (Share); cashflow ratio (Cashflow); return on assets (ROA); γ is a time-fixed effect; μ is an individual fixed effect; and σ is an error term.

### 3.3 Variable description

Explained variables: Two common indicators for measuring firm innovation are generally used: one is the number of patents, including applications, grants and citations of patents [35],

**Table 1. Sample enterprises of new energy vehicles.**

| Industry chain division | Enterprise name | | |
|---|---|---|---|
| Upstream enterprises (mainly including lithium ore, cobalt ore, nickel ore, rare earths and other raw material development and processing enterprises) | Tianqi Lithium Corporation | Ganfeng Lithium Group Co., Ltd | Zhejiang Huayou Cobalt Co., Ltd. |
| | Fangda Carbon New Material Co., Ltd | Guangzhou Tinci Materials Technology Co., Ltd. | Shanghai Putailai New Energy Technology Co., Ltd. |
| | Tonze New Energy Technology Co., Ltd. | Shanxi Meijin Energy Co., Ltd. | Shenzhen Dynanonic Co., Ltd. |
| | Shenzhen Capchem Technology Co., Ltd | | |
| Midstream enterprises (mainly including power battery, motor, electric control, positive and negative electrode materials, diaphragm, electrolyte production and manufacturing enterprises) | Contemporary Amperex Technology Co., Limited | Gotion High-tech Co., Ltd. | Sunwoda Electronic Co., Ltd. |
| | Guangzhou Great Power Energy and Technology Co., Ltd. | Weichai Power Co., Ltd. | Han's Laser Technology Industry Group Co., Ltd. |
| | Hubei Kailong Chemical Group Co., Ltd. | Nanjng YueBoo Power System Co., Ltd. | Zhengzhou Tiamaes Technology Co., Ltd. |
| | BAIC BluePark New Energy Technology Co., Ltd. | Tianjin Motor Dies Co., Ltd. | Zhongshan Broad-Ocean Motor Co., Ltd. |
| | Zhejiang Founder Motor Limited Company | Jiangxi Special Electric Motor Co., Ltd. | Wanxiang Qianchao Co., Ltd. |
| | Zhejiang Asia-Pacific Mechanical and Electronic Company Limited | | |
| Downstream enterprises (enterprises with R&D, manufacturing, assembly and sales of new energy vehicles) | BYD Company Limited | Yutong Bus Co., Ltd | Anhui Jianghuai Automobile Group Corp., Ltd. |
| | FAW Jiefang Group Co., Ltd | Yangzhou Yaxing Motor Coach Co., Ltd. | Zhongtong Bus Holding Co., Ltd |
| | Anhui Ankai Automobile Co., Ltd. | Beiqi Foton Motor Co., Ltd. | SAIC Motor Corporation Limited |
| | Haima Automobile Co., Ltd | Dongfeng Automobile Co., Ltd. | Jiangling Motors Corporation, Ltd. |
| | Xiamen King Long Motor Group Co., Ltd. | Great Wall Motor Company Limited | Lifan Technology (Group) Co., Ltd. |
| | Liaoning SG Automotive Group Co., Ltd. | Shenyang Jinbei Automotive Company Limited | China Railway Materials Company Limited |
| | Bohai Automotive Systems Co., Ltd. | CRRC Corporation Limited | China Grand Automotive Services Group Co., Ltd. |
| | Sinotruk Jinan Truck Co., Ltd. | Guangzhou Automobile Group Co., Ltd. | Seres Group Co., Ltd. |

while the other is the use of sales revenue from new products [36]. As the sales revenue of new products has not been published in the annual reports of enterprises, based on the availability of data, this study uses the number of patent applications to indicate the innovation capability of enterprises, which includes invention patents, utility model patents and design patents, and searches the patent applications of sample enterprises on new energy vehicles on the pages of the State Intellectual Property Office of China (SIPO) and the 2010 World Intellectual Property Organization provided. The number of new energy vehicle-related patents filed by enterprises each year is obtained by matching the IPC classification numbers of new energy vehicle-related patents with the types of enterprise-level patents obtained from the SIPO search.

Core explanatory variables: Government subsidies are usually divided into direct and indirect subsidies. The former usually refers to government subsidies for enterprises' innovation

**Table 2. Variable descriptions.**

| Variable Name | Variable Symbol | Variable Explanation | Data source |
|---|---|---|---|
| Number of new energy vehicle patent applications | Patent | The number of patents applied for by enterprises in the year plus one to take the logarithm, unit (pieces) | SIPO |
| Government subsidies | Subsidy | Substitute and take the logarithm of government grants included in current profit or loss in the annual report, unit (CNY) | Manual collation from corporate annual reports |
| R&D expenditure | Expenditure | R&D investment reported in the annual report and taken as a logarithm, unit (CNY) | Manual collation from corporate annual reports |
| R&D staff input | Staff | Number of R&D personnel reported in the annual report and taken as a logarithm, unit (person) | Manual collation from corporate annual reports |
| Age of the enterprise | Age | Expressed as the year in which the enterprise is located in the current period minus the year of establishment, in unit (year) | Manual collation from corporate annual reports |
| Asset and liability ratio | Debt | Total corporate liabilities / total corporate assets, unit (%) | CSMAR Database |
| Tobin Q | Tobin Q | Market capitalization of enterprises/total assets, unit (%) | CSMAR Database |
| Shareholding ratio of the first largest shareholder | Share | Shareholding of the first largest shareholder of the enterprise, unit (%) | CSMAR Database |
| Cashflow ratio | Cashflow | Expressed as net cash flows from operating activities of the enterprise for the period divided by total assets, unit (%) | WIND Database |
| Return on assets | ROA | Replaced by the weighted average return on net assets in the annual report unit (%) | WIND Database |

activities, while the latter usually connotes tax rebates, tax incentives, and other relief measures for enterprises [37]. As the annual reports of Chinese A-share companies do not disclose data related to direct government subsidies, the "government subsidies included in current profit and loss" in the annual reports are used as a proxy variable, which may be greater than the actual amount of direct government subsidies, but the difference is not too large [38].

Control variables: The number of enterprise R&D personnel and enterprise R&D expenditure can respond to the scale of enterprise R&D investment, and the intensity of enterprise R&D innovation activities will increase with the scale of enterprise R&D investment [39]. Enterprises with larger R&D scales may have achieved economies of scale with better R&D innovation capability. They are more likely to be favored by the government and receive more subsidies compared to enterprises with smaller R&D investments [40]. Older firms are more likely to receive subsidies compared to those with lower years of establishment [41], while younger firms do not have a significant advantage in receiving subsidies. Innovation activities are characterized as long-term and high-risk for enterprises. Moreover, when the financial situation of enterprises is poor, the innovation behavior of enterprises will be significantly affected, while corporate governance can influence the innovation activities of enterprises by affecting the enthusiasm of employees, and the concentration of equity can better reflect the governance of enterprises [42], represented by the shareholding ratio of the first largest shareholder in this study. The fundamental motive of corporate innovation is to obtain excess profit and higher return on capital, and the profitability condition and financial status of the company itself greatly determine the innovative behavior of the company. Thus, this study controls for the liability ratio, Tobin's Q, return on assets, and cashflow ratio.

Table 2 presents the measures and sources of each variable. To reduce noise from extreme data, all data are winsorized at the 1% and 99% quartiles.

## 3.4 Variable data analysis

Table 3 shows the descriptive statistics of each variable (some variables have been logarithmically processed). The patent output of new energy vehicle enterprises varies from 0.00 to 8.503,

**Table 3. Descriptive statistics.**

| VarName | Obs | Mean | SD | Skewness | Min | Median | Max |
|---|---|---|---|---|---|---|---|
| Patent | 355 | 4.501 | 2.054 | -0.047 | 0.000 | 4.331 | 8.503 |
| Subsidy | 355 | 18.047 | 1.771 | 0.108 | 13.407 | 17.853 | 21.848 |
| Subsidy$^2$ | 355 | 328.814 | 64.383 | 0.352 | 179.749 | 318.741 | 477.327 |
| Expenditure | 351 | 19.539 | 1.783 | 0.161 | 13.969 | 19.402 | 23.318 |
| Staff | 355 | 6.733 | 1.701 | 0.230 | 2.079 | 6.446 | 10.471 |
| Age | 355 | 3.025 | 0.441 | -0.337 | 1.609 | 3.045 | 4.094 |
| Debt | 355 | 0.570 | 0.187 | -0.270 | 0.104 | 0.571 | 0.973 |
| Tobin Q | 355 | 1.752 | 0.996 | 2.199 | 0.757 | 1.443 | 6.282 |
| Share | 354 | 0.345 | 0.161 | 0.709 | 0.053 | 0.301 | 0.773 |
| Cashflow | 355 | 0.039 | 0.080 | -1.067 | -0.305 | 0.042 | 0.238 |
| ROA | 355 | 0.043 | 0.259 | -3.028 | -1.245 | 0.077 | 0.466 |

with a mean value of 4.501 and a standard deviation of 2.054, indicating large differences between the innovation outputs of different new energy vehicle enterprises and that the sample is informative. Additionally, the mean value of government subsidies is 18.047 with a standard deviation of 1.771, and the overall representativeness of the sample is good. To avoid pseudo-regression, the correlation coefficient test is conducted for each variable, and the correlation between most of the variables is significant at the 1% level, with a high degree of correlation, which reduces the possibility of pseudo-regression to a greater extent. In addition, all the variables pass the variance inflation factor test, and the coefficients of each are much less than 10, indicating absence of multicollinearity among the variables, which provides the basis for the subsequent empirical study.

# 4. Empirical results

## 4.1 Basic regression analysis

Panel regression is used to empirically test 50 new energy vehicle companies, and both static and dynamic panels are used for estimation to ensure the robustness of the results. All regressions are subjected to the Hausman test before running a regression, and the results show that fixed effects should be selected for estimation. Furthermore, all the AR(2) values in dynamic panel estimation are greater than 0.1 and pass the Sargon test to ensure the accuracy of dynamic panel estimation. Two methods, namely, differential and systematic gaussian mixture model (GMM), are adopted for dynamic panel estimation. The results of the underlying regression are shown in Table 4. Columns (1), (3), and (4) verify the linear mechanism of government subsidies on the innovation of new energy vehicle enterprises. This indicates a significant promotion effect of government subsidies on the innovation of new energy vehicle enterprises, while the dynamic panel estimation shows that the innovation output of enterprises in the previous period would significantly increase the innovation output in the current period, thereby verifying the inertia of enterprise innovation [43, 44]. Columns (2), (5), and (6) further investigate the nonlinear mechanism of government subsidies based on linear estimation, and the results show that the coefficients of Subsidy$^2$ are significantly negative at a significance level greater than 5%, indicating that there is a certain crowding-out effect of government subsidies on enterprise innovation. There is an inverted "U" shaped relationship, which is consistent with the findings of Dai, Liu et al. [45, 46] thus, H1 is verified. Further analysis shows that the inflection point of the inverted U curve of government subsidies is 18.593, while the average value of government subsidies for the sample enterprises is 18.047, which

**Table 4. Basic regression.**

| | Linear static panel estimation | Nonlinear static panel estimation | Linear dynamic panel estimation | | Nonlinear dynamic panel estimation | |
|---|---|---|---|---|---|---|
| | | | systematic GMM | differential GMM | systematic GMM | differential GMM |
| | (1) | (2) | (3) | (4) | (5) | (6) |
| Subsidy | 0.116* | 1.599** | 0.180* | 0.106 | 1.907*** | 3.833*** |
| | (1.858) | (2.537) | (1.903) | (1.338) | (2.630) | (4.807) |
| Subsidy$^2$ | | -0.043** | | | -0.047** | -0.107*** |
| | | (-2.365) | | | (-2.276) | (-4.716) |
| Expenditure | 0.487*** | 0.477*** | 0.482*** | 0.341*** | 0.536*** | 0.472*** |
| | (4.895) | (4.825) | (5.176) | (2.736) | (4.080) | (4.227) |
| Staff | -0.086 | -0.075 | -0.217** | -0.172 | -0.312** | -0.124 |
| | (-0.765) | (-0.671) | (-2.423) | (-1.416) | (-2.342) | (-1.068) |
| Age | -2.453** | -2.269** | 1.361*** | -0.097 | 1.437*** | -5.596*** |
| | (-2.388) | (-2.220) | (6.628) | (-0.213) | (5.608) | (-4.240) |
| Debt | -1.034* | -1.245** | -3.152*** | -1.031 | -3.651*** | -2.520*** |
| | (-1.867) | (-2.236) | (-5.373) | (-1.372) | (-4.482) | (-3.301) |
| Tobin Q | 0.030 | 0.039 | 0.392*** | -0.149** | 0.283*** | 0.229** |
| | (0.439) | (0.575) | (5.039) | (-2.025) | (2.843) | (2.506) |
| Share | 3.782*** | 4.059*** | 1.846* | 2.384** | 2.069* | 2.332** |
| | (3.842) | (4.127) | (1.696) | (2.297) | (1.812) | (2.225) |
| Cashflow | -0.469 | -0.817 | 0.615 | -0.529 | 0.758 | -2.027** |
| | (-0.657) | (-1.132) | (0.703) | (-0.681) | (0.807) | (-2.575) |
| ROA | -0.054 | -0.021 | -0.088 | -0.319 | -0.130 | -0.188 |
| | (-0.276) | (-0.111) | (-0.577) | (-1.548) | (-0.584) | (-0.967) |
| Patent$_{t-1}$ | | | 0.461*** | 0.255*** | 0.467*** | 0.152*** |
| | | | (9.549) | (4.398) | (7.985) | (2.678) |
| _cons | Yes | Yes | Yes | Yes | Yes | Yes |
| Year | Yes | Yes | Yes | Yes | Yes | Yes |
| Id | Yes | Yes | Yes | Yes | Yes | Yes |
| N | 350 | 350 | 347 | 297 | 347 | 297 |
| R$^2$ | 0.301 | 0.315 | | | | |

Notes: t values are reported in parentheses.

*p<0.1,

**p<0.05,

***p<0.01

indicates that the current government subsidies are basically at the inflection point of the inverted U curve vis-à-vis the promotion of innovation in new energy vehicle enterprises.

## 4.2 Robustness tests

There may be a strong endogeneity problem between government subsidies and enterprise innovation, and the more innovative enterprises may receive more government subsidies. To ensure the reliability of the regression results and to avoid the endogeneity problem, this study also conducts the following robustness tests.

Considering the possible long time lag of government subsidies, the explanatory variable patent is regressed again with one period lag to test the robustness of the findings, and the results are shown in Table 5(1), where the coefficient of the core explanatory variable Subsidy$^2$

**Table 5. Robustness tests.**

| | Dependent variable lags one period (1) | Instrumental Variables (2) | Replace the dependent variable (3) | Replace the control variable (4) |
|---|---|---|---|---|
| Subsidy | 1.559** | 10.730** | 0.019** | 2.195*** |
| | (2.283) | (2.255) | (2.269) | (3.076) |
| Subsidy$^2$ | -0.039** | -0.306** | -0.000** | -0.059*** |
| | (-2.034) | (-2.240) | (-2.043) | (-2.877) |
| Expenditure | 0.472*** | 0.430*** | | |
| | (4.779) | (3.011) | | |
| Staff | 0.011 | 0.018 | 0.005*** | |
| | (0.100) | (0.106) | (3.682) | |
| Age | -1.703* | -1.237 | 0.016 | -1.389 |
| | (-1.684) | (-0.967) | (1.213) | (-1.156) |
| Debt | -1.056* | -2.403** | -0.009 | 0.055 |
| | (-1.921) | (-2.549) | (-1.267) | (0.079) |
| Tobin Q | -0.164** | 0.087 | -0.001 | -0.082 |
| | (-2.420) | (1.105) | (-1.071) | (-0.994) |
| Share | 3.950*** | 6.020*** | -0.012 | 4.207*** |
| | (4.006) | (3.793) | (-0.955) | (3.788) |
| Cashflow | -0.191 | -2.877** | -0.002 | -0.974 |
| | (-0.266) | (-2.062) | (-0.212) | (-1.134) |
| ROA | -0.064 | 0.139 | 0.000 | -0.334 |
| | (-0.336) | (0.552) | (0.179) | (-0.655) |
| Profit | | | | 0.085 |
| | | | | (1.457) |
| Total staff | | | | 0.374* |
| | | | | (1.934) |
| _cons | Yes | Yes | Yes | Yes |
| Year | Yes | Yes | Yes | Yes |
| Id | Yes | Yes | Yes | Yes |
| N | 347 | 349 | 350 | 288 |
| R$^2$ | 0.406 | 0.821 | 0.168 | 0.287 |

Notes: t values are reported in parentheses.

*p<0.1,

**p<0.05,

***p<0.01

is -0.039, which is significantly negative at the 5% significance level, again confirming the inverted U-type relationship between government subsidies and corporate innovation.

To overcome the endogeneity in sample selection, the two-stage least squares method is used for instrumental variable (IV) testing, aiming to effectively circumvent the possible endogeneity problem. Government subsidies with a one-period lag are selected as the IVs, which usually have an impact on current corporate innovation, while current corporate innovation did not have an impact on government subsidies in the previous period, and the use of this IV can avoid the endogeneity problem to some extent. The IVs all pass the weak IV and non-identifiable tests, and the regression results are shown in Table 5(2), where the coefficient of Subsidy$^2$ is -0.306, which is significantly negative at the 5% significance level.

Corporate innovation is divided into two processes, namely, input and output, and can be measured from both the input and output perspectives. In this section, the number of patent

applications from the output perspective is replaced with the RD input intensity, which can reflect the size of corporate innovation capability [47]. The measure is corporate R&D investment divided by total corporate assets, and the regression results are shown in Table 5(3), where the coefficient of Subsidy$^2$ is significantly negative at the 5% significance level and the conclusion remains robust.

Replacing the control variables, R&D personnel and R&D expense input are the key variables that directly affect the innovation output of enterprises. They may be highly correlated with the innovation output of enterprises; the higher the profits of enterprises, the more they may invest in R&D expenses; and the total number of employees can also reflect the number of R&D personnel on the side. Replacing the R&D expenses and R&D personnel input with operating profit and the total number of employees in service regression again. The results are shown in Table 5(4), and the coefficient of Subsidy$^2$ is -0.059, which is significantly negative at the 1% significance level. The results remain consistent with the previous conclusion.

## 5. Heterogeneity analysis

### 5.1 Enterprise heterogeneity analysis

The samples of new energy vehicle enterprises are divided into state- and non-state-owned new energy vehicle enterprises. The division is based on the ownership of the first largest shareholder, defined as an SOE if the first largest shareholder is a state-owned legal person and a non-SOE if it is a non-state-owned legal person. The results of the heterogeneity of property rights are shown in Table 5(1)(2). The Subsidy$^2$ coefficient of non-SOEs is -0.043 (p-value <0.1), while that of SOEs is -0.035 (p-value >0.1), which indicates that the incentive effect of government subsidies is more prominent for non-SOEs than SOEs. The inverted U curve is steeper, and the incentive effect of government subsidies on SOEs is not statistically significant. This is possibly because SOEs will have the underwriting guarantee from the government, and they may form a dependence effect on innovation incentives due to this underwriting guarantee and will not be eliminated from the market due to the decline of innovation ability. However, non-SOEs must continuously innovate their products to adapt to the severe market competition due to their self-financing characteristics, and the characteristics of the market economy determine the inevitability of innovation incentives for non-SOEs.

According to the main businesses of the sample enterprises, they are divided into upstream, midstream, and downstream industrial chains (see Table 1), and the results of industrial chain heterogeneity are shown in Table 5(3)(4)(5). The coefficients of downstream vehicle enterprises, midstream component enterprises, and upstream raw material Subsidy$^2$ are -0.053 (p-value <0.05), 0.026 (p-value >0.1), and 0.001 (p-value >0.1), respectively. This indicates that government subsidies significantly stimulate the innovation activities of vehicle enterprises, while the innovation incentives of component enterprises and raw material enterprises are not significant. In addition, the trend of the inverted U curve for downstream vehicle enterprises is significant, while that for component enterprises and raw material enterprises is not significant. This is because the value-added products of downstream vehicle enterprises are higher, their innovation power will directly affect the market, and the feedback effect of the market on vehicle innovation is greater, which is more likely to promote the public's interest in and consumption of new energy vehicles. Furthermore, this is more conducive to the prosperity of the new energy vehicle market, which helps expand the consumer market and is more likely to be favored by the government, thus obtaining more subsidies and forming a virtuous circle. Conversely, component companies and raw material companies may not be able to play a direct role in the market because they are at the middle

**Table 6. Enterprise heterogeneity.**

| | SOEs (1) | Non-SOEs (2) | Downstream enterprises (3) | Midstream enterprises (4) | Upstream enterprises (5) | More than 20 years (6) | Less than 20 years (7) |
|---|---|---|---|---|---|---|---|
| Subsidy | 1.381 | 1.486* | 1.952** | -0.890 | 0.446 | 1.094* | 8.322*** |
| | (1.329) | (1.710) | (2.054) | (-0.524) | (0.193) | (1.658) | (2.852) |
| $Subsidy^2$ | -0.035 | -0.043* | -0.053** | 0.026 | 0.001 | -0.028 | 0.230*** |
| | (1.199) | (-1.740) | (-1.978) | (0.550) | (0.022) | (-1.457) | (-2.823) |
| Expenditure | 0.471** | 0.530*** | 0.716*** | -0.145 | 0.334 | 0.496*** | 0.555 |
| | (2.311) | (4.015) | (4.739) | (-0.505) | (1.440) | (4.585) | (1.646) |
| Staff | -0.006 | 0.057 | -0.182 | 0.838*** | -0.072 | -0.139 | 0.094 |
| | (0.031) | (0.376) | (-1.047) | (2.652) | (0.332) | (-1.080) | (0.307) |
| Age | -1.314 | -2.135* | -3.858** | -2.175 | 2.628 | -3.242* | -0.434 |
| | (0.579) | (-1.757) | (-2.057) | (-1.078) | (1.207) | (-1.928) | (-0.174) |
| Debt | 2.330** | -0.195 | -1.293 | 0.464 | -2.230 | -0.862 | -4.079** |
| | (2.123) | (-0.262) | (-1.477) | (0.466) | (1.643) | (-1.377) | (-2.142) |
| Tobin Q | 0.335 | -0.037 | 0.283 | -0.194 | 0.065 | 0.043 | 0.048 |
| | (1.303) | (-0.448) | (1.570) | (-1.568) | (0.459) | (0.515) | (0.282) |
| Share | 3.511** | 2.497* | 4.213*** | 2.089 | 2.024 | 4.152*** | 5.196* |
| | (2.251) | (1.707) | (3.193) | (0.808) | (0.512) | (3.726) | (1.779) |
| Cashflow | 0.002 | 2.191** | -0.253 | 1.662 | -2.323 | -0.148 | 6.196** |
| | (0.002) | (-2.209) | (-0.249) | (0.963) | (1.285) | (-0.187) | (-2.585) |
| ROA | -0.085 | 0.429 | -0.075 | 0.169 | 0.488 | -0.076 | 0.037 |
| | (-0.313) | (1.255) | (-0.293) | (0.371) | (0.663) | (-0.354) | (0.067) |
| _cons | Yes | Yes | Yes | Yes | Yes | Yes | Yes |
| Year | Yes | Yes | Yes | Yes | Yes | Yes | Yes |
| Id | Yes | Yes | Yes | Yes | Yes | Yes | Yes |
| N | 129 | 220 | 184 | 103 | 63 | 283 | 64 |
| $R^2$ | 0.438 | 0.350 | 0.337 | 0.525 | 0.491 | 0.303 | 0.574 |

Notes: t values are reported in parentheses.

*$p<0.1$,

**$p<0.05$,

***$p<0.01$

and upper ends of the industry chain, so they have fewer chances to obtain government subsidies and fewer incentives to innovate.

To examine the variability of the years of establishment, this study groups the sample enterprises according to the actual situation of the development of new energy vehicle enterprises and previous research results [48], using 20 years of establishment as the cut-off point, and the difference in the years of establishment of the enterprises is shown in Table 6(6)(7). The coefficients of $Subsidy^2$ for enterprises with more than 20 years of establishment and those with less than 20 years of establishment are -0.028 (p-value>0.1) and -0.230 (p-value<0.01), respectively, indicating that government subsidies are more significant for enterprises with fewer years of establishment and that the inverted U curve tends to be significant. The government subsidies provide these young enterprises with the opportunity to innovate and make trial-and-error decisions, while the older enterprises may have formed a monopoly in the market, so the incentive to innovate is insufficient and that the government subsidies can hardly stimulate their innovation activities. After the above analysis, H2 is verified.

## 5.2 Regional heterogeneity analysis

Referring to Han [49], the sample firms are classified into eastern and non-eastern firms based on the address of each firm's registered office and the provincial administrative divisions of China (eastern regions include: Hebei Province, Beijing, Tianjin, Shandong Province, Jiangsu Province, Shanghai, Zhejiang Province, Fujian Province, Guangdong Province, and Hainan Province). The regional differences are presented in Table 6(1)(2). The coefficients of $Subsidy^2$ for eastern and non-eastern enterprises are 0.006 (p-value>0.1) and -0.078 (p-value<0.1), respectively, indicating that government subsidies are more significant for non-eastern enterprises. Additionally, the inverted U curve tends to be significant, probably because of the higher level of economic development in the eastern region, and the technology level of enterprises in the eastern region is also relatively higher. Thus, the policy incentive effect of enterprises in the eastern region is lesser than that of enterprises in the non-eastern region, while the marginal benefit of the incentive effect of subsidies is greater in the non-eastern region due to the backwardness of resources and technology. Furthermore, in the eastern region, the economic development level is generally higher, as is the policy implementation level, and the transparency of government departments is also higher than that of the non-eastern region.

Referring to Chen [50], to avoid endogeneity, the ratio of keywords about environmental issues in the work report of the government of the district where the new energy vehicle enterprises are located to the total number of words in the report is taken to measure the intensity of environmental regulation. The keyword extraction tool is Python. The sample enterprises are divided into enterprises in high environmental regulation areas and enterprises in low environmental regulation areas according to the median of the intensity of environmental regulation. The difference in environmental regulation is shown in Table 7(3)(4). The coefficients of $Subsidy^2$ are -0.047 (p-value < 0.05) and -0.081 (p-value < 0.05), respectively, for firms in high and low environmental regulation areas, compared to government subsidies for firms in low environmental regulation areas, and the inverted U curve tends to be significant. The possible reason for this is that the marginal benefit of the subsidy incentive is greater in areas with low environmental regulations, where the development of new energy vehicles is slow due to technology and resource constraints. Thus, H3 is verified.

## 6. Conclusion

This study empirically analyzes the relationship between government subsidies and innovation of new energy vehicle enterprises, taking 50 listed new energy vehicle enterprises in Shanghai and Shenzhen A-shares from 2012 to 2021 as a sample. It is found that, first, government subsidies promote the innovation of new energy vehicle enterprises to a certain extent, but beyond a certain limit, they will play a suppressive role, that is, there is an inverted U-type relationship. The current level of subsidies has reached the inflection point of the inverted U curve, and the stimulating effect of government subsidies has reached the threshold. Second, the incentive effect of government subsidies on the innovation of new energy vehicle enterprises is heterogeneous at the enterprise and regional levels. The incentive effect of government subsidies on the innovation of non-SOEs, enterprises in the downstream of the industry chain, and enterprises with lower establishment years is significant. Moreover, the trend of an inverted U-type relationship is evident, while the incentive effect on SOEs, enterprises in the middle and upper streams of the industry chain and enterprises with higher growth years is not significant, and the inverted U-type relationship is not evident. Third, the promotion effect of government subsidies on enterprises in non-eastern regions is significant, and the trend of an inverted U-type relationship is evident, while that in eastern regions is not significant. The inverted

**Table 7. Regional heterogeneity.**

|  | Eastern (1) | Non-eastern (2) | High environmental regulation (3) | Low environmental regulation (4) |
|---|---|---|---|---|
| Subsidy | 0.001 | 2.563* | 1.940** | 2.678** |
|  | (0.002) | (1.689) | (2.292) | (2.249) |
| Subsidy$^2$ | 0.006 | -0.078* | -0.047** | -0.081** |
|  | (0.340) | (-1.821) | (-1.974) | (-2.385) |
| Expenditure | 0.112 | 0.701*** | 0.323* | 0.809*** |
|  | (0.933) | (3.681) | (1.926) | (5.223) |
| Staff | 0.078 | 0.038 | -0.301 | -0.065 |
|  | (0.585) | (0.189) | (-1.397) | (-0.412) |
| Age | 0.233 | 1.881** | -2.627 | -2.530 |
|  | (0.447) | (2.029) | (-1.439) | (-1.490) |
| Debt | 2.353*** | 0.492 | 3.011*** | 0.691 |
|  | (-3.070) | (0.514) | (-3.093) | (0.885) |
| Tobin Q | -0.083 | -0.197 | 0.102 | 0.040 |
|  | (-1.390) | (-1.346) | (0.932) | (0.326) |
| Share | 1.093 | 4.899*** | 3.669*** | 4.266** |
|  | (0.613) | (3.387) | (2.842) | (2.078) |
| Cashflow | -0.637 | 0.767 | -1.143 | 0.668 |
|  | (-0.850) | (0.505) | (-1.045) | (0.614) |
| ROA | -0.429** | 0.302 | 0.038 | -0.005 |
|  | (-2.018) | (0.798) | (0.115) | (-0.020) |
| _cons | Yes | Yes | Yes | Yes |
| Year | Yes | Yes | Yes | Yes |
| Id | Yes | Yes | Yes | Yes |
| N | 220 | 130 | 171 | 179 |
| R$^2$ | 0.185 | 0.370 | 0.451 | 0.356 |

U-shaped relationship trend is more pronounced in low environmental regulation areas than in high ones, where government subsidies contribute more significantly.

Our study also provides two policy implications. First, improve the subsidy policy for new energy vehicle enterprises. In the general environment of the gradual retreat of new energy vehicle subsidies, the government's role in regulating and leading the process of changing new energy vehicles from government-led to market-led is indispensable, and the subsidies should be gradually and gently retracted during the retreat stage [51]. Simultaneously, improve the threshold for new energy vehicle enterprises to apply for government subsidies, reduce subsidies while achieving optimization and upgrading of the new energy vehicle industry, improve the technical capabilities of new energy vehicle enterprises, increase the supervision of new energy vehicle enterprises applying for government subsidies, and avoid wrong subsidies and abusive subsidies to prevent "bottom-up competition" in the new energy vehicle market.

Second, develop a differentiated government subsidy system. At present, China's new energy vehicle industry is at a critical market-led development stage. For the government, blindly providing subsidies will not only mean high policy costs but also reduce the efficiency of enterprise innovation. Therefore, in the process of providing subsidies, they should be targeted and clearly defined, reflecting a differentiated strategy for different regions and the nature of enterprises. To accelerate the high-quality development of new energy vehicles, the Chinese government should focus its subsidy level and policy regulation on SOEs, midstream

and upstream enterprises, as well as enterprises in non-eastern regions with lower environmental regulation intensity. It should also allocate limited resources to such enterprises to improve the efficiency of subsidies and accelerate the transformation and upgrading of the new energy vehicle industry.

## Supporting information

**S1 Data.**
(DTA)

## Author Contributions

**Conceptualization:** Jianguo Sun.

**Formal analysis:** Weitong Zhang.

**Investigation:** Mingfu Tian.

**Methodology:** Mingfu Tian.

**Visualization:** Jingyi Ning.

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
