## [Decision Letter · Decision Letter 0]

18 Jan 2023

PONE-D-22-30783Government subsidies and innovation in new energy vehicle companies--Empirical study of new energy vehicle listed companies based on Shanghai and Shenzhen A-sharesPLOS ONE

Dear Dr. tian,

Thank you for submitting your manuscript to PLOS ONE. After careful consideration, we feel that it has merit but does not fully meet PLOS ONE’s publication criteria as it currently stands. Therefore, we invite you to submit a revised version of the manuscript that addresses the points raised during the review process.

We look forward to receiving your revised manuscript.

Kind regards,

Hung Do

Academic Editor

PLOS ONE

Journal Requirements:

4. Please amend your authorship list in your manuscript file to include author Jingyi Ning and Weitong Zhang.

5. Please amend the manuscript submission data (via Edit Submission) to include author WeiZhong Zhang.

Reviewers' comments:

Reviewer's Responses to Questions

**Comments to the Author**

1. Is the manuscript technically sound, and do the data support the conclusions?

Reviewer #1: Partly

Reviewer #2: No

2. Has the statistical analysis been performed appropriately and rigorously? 

Reviewer #1: No

Reviewer #2: No

3. Have the authors made all data underlying the findings in their manuscript fully available?

Reviewer #1: No

Reviewer #2: Yes

4. Is the manuscript presented in an intelligible fashion and written in standard English?

Reviewer #1: No

Reviewer #2: No

5. Review Comments to the Author

Reviewer #1: Refer to the attached file.

Reviewer’s Report on Manuscript PONE-D-22-30783 titled “Government subsidies and innovation in new energy vehicle companies: Empirical study of new energy vehicle listed companies based on Shanghai and Shenzhen A-shares”

This paper studies the effect of government subsidies on corporate innovation of new energy vehicle enterprises (NEV) in China using panel regression model. The results show that the effects of government subsidies on NEV innovation is non-linear and heterogeneous depending on the role of firms in the value chain of industry, the firm age and type of firms. Particularly, the innovation effect is significant for downstream vehicle enterprises, but no such effect for midstream core component and upstream raw material enterprises. Further, the innovation effect of subsidies is more pronounced in young enterprises compared to old enterprises. In addition, the subsidies only induce significant effects on innovation of non-state-owned firms while the innovations of state-owned firms are not affected by government subsidies.

The NEV has recently emerged as the strategic industry in China’s Industry Development Plan to achieve the country’s low-carbon emission and sustainable energy consumption targets, thereby the literature on the determinants of corporate innovation in China NEV industry has gained extensive attention from both researchers and practitioners. To my knowledge, there are abundant studies investigating the innovation effect of government subsidies on NEV firms, including Jiang et al. (2018), Li et al. (2021), Shao et al. (2021), to name a few. These papers all reveal that there is non-linear relationship between government subsidies and innovation in NEV firms in China. Further, the innovation effect intensity is region-dependent, varies by the type of subsidies and the type of enterprises.

This paper aims to extend the literature by providing the evidence of an inverted U-shaped relationship between government subsidies on NEV firm innovation in China. However, in the introduction section, the authors only state the research subject which is investigating the effect of government subsidies on firm innovation. Thereby, they failed to differentiate their study from the existing studies in term of research issue. That is, the paper limits its academic contribution. Further, the other main contribution of the paper is investigating how the effect of government subsidies on firm innovation changes depending on the role of enterprises in value chain of the industry, the type of firms (state-owned firms vs non-state-owned firms) and the firm age. So, in the current introduction section, the authors should provide appropriate arguments and explanations to highlight this main contribution of the paper. The study also needs more in-depth analyses regarding methodology, variable selection and empirical analyses.

So, I have several comments/ suggestions to improve the quality as well as the contribution of the paper, as follows:

The current introduction seems to provide too much background information regarding China NEV industry while the motivation and contribution are not made clear. I suggest the following:

Shorten the background information, brief review of the relation between government subsidies and firm innovation in general (i.e., existing evidence of non-linear relationship, crowding-out effect in this relationship),

State the research subject of the paper in the first/ second paragraph of the introduction which is different with the existing studies as it first provides the evidence of inverted U-shape relationship on NEV firm innovation in China.

State the reason why it is important to explore how the effect of government subsidies varies by the role of firms in value chain, the firm type, and the firm age. This would help highlight the main motivation and contribution of the paper in introduction section.

In literature review, the paper provides appropriate hypothesis developments. However, regarding the hypothesis 1, I suggest citing more relevant and recent literature as mentioned above. Please check the duplicate sentences (which happen quite frequently) and redraft the literature review section.

Regarding variable selection, I perceive that there are many wrong descriptions of the variables used in the model. For example, “Patent¬¬it is the explanatory variable”??? More importantly, the authors did not provide the reasons why they only use Patent as a proxy for firm innovation. Since the major existing study documents that firm innovation is an input-output process which could be proxied by R&D spending and the number of the patents. (Li et al., 2021). On this matter, I would prefer to see more detailed explanation on the selection of the dependent variable. Turning to the selection of control variables, the paper should cite more relevant papers which document the correlations between there control variables with firm innovation. There is a lack of description of ¬¬¬¬t and μ_i variables in the model (Are they denotations of year fixed effect and firm fixed effect, respectively?)

Methodology: Existing studies have extensively recognized that there are potential endogeneity issues in the regression of firm innovation on subsidies. That means the subsidy is the possible endogenous explanatory variable in the regression (Clausen, 2009 and Li et al., 2021). To address the potential endogeneity issues, I suggest conducting the Difference-in-Difference (DiD) test first, then the authors can run the Difference GMM and System GMM to estimate the model instead of using the static regression as the main methodology. As I perceive that the authors have employed GMM models in robustness checking, I suggest moving this empirical result to the main analyses and citing more supporting literature on the potential endogeneity issue of subsidy variable (the current paper does not cite the relevant studies regarding the issue of endogenous explanatory variable)

Analyses of empirical results: The authors do not link the results of the paper to relevant findings in the extant literature. As such, the paper’s results remain isolated and then its contribution is limited.

Policy implications: The current content of policy implication section is unlinked to the paper’s findings and somewhat irrelevant to the study. This way, the paper’s results do not provide insights for policy makers. I would prefer to see each policy implication is directly linked to the finding of the paper.

Minor comment

The writing is often vague. You need to be as succinct as possible. You should avoid writing such a lengthy and duplicate sentence, as follows “This paper analyzes the mechanism of government subsidies on the innovation of new energy vehicle enterprises and the differences in the impact of different types of enterprises from a quantitative perspective, based on the panel data of new energy vehicle enterprises from 2012-2021, under the micro subject of new energy vehicle enterprises, which enriches the relevant research on new energy vehicle micro enterprises and is of great value to the policy formulation and stimulation of enterprise innovation of new energy vehicles in China. It has important reference value for the policy formulation and stimulation of enterprise innovation in China.”. I suggest dropping the duplicate sentence, breaking down the lengthy sentence to shorter ones. The paper needs to be redrafted and polished thoroughly to improve the conciseness of the paper.

The current abstract also needs to be rewritten to remove some duplicates. Again, please avoid using many long-winded sentences (the sentences include many clauses connected by semi-colon) to present the findings in the abstract.

Please check for grammatical mistakes and sentence structure as many writing errors appear across the paper. For example, “Differences in property rights, industry chain division of labor, and the number of years an enterprise has been established have different incentives for innovation in government subsidies, which requires "teaching different types of enterprises according to their abilities", giving full play to the advantages of China's state-owned enterprises, while improving the innovative vitality of non-stateowned enterprises; improving the adaptability of the new energy vehicle industry chain division of labor, and setting different incentives for enterprises in different industry chains”. The writing seems to be tedious and needs to be enhanced a lot to meet the standard of an academic research.

Please cross-check the reference list and citations to make sure that all citations are listed in the reference.

Reference

Clausen, T. H. (2009). Do subsidies have positive impacts on R&D and innovation activities at the firm level?. Structural change and economic dynamics, 20(4), 239-253.

Jiang, C., Zhang, Y., Bu, M., & Liu, W. (2018). The effectiveness of government subsidies on manufacturing innovation: Evidence from the new energy vehicle industry in China. Sustainability, 10(6), 1692.

Li, Q., Wang, M., & Xiangli, L. (2021). Do government subsidies promote new-energy firms’ innovation? Evidence from dynamic and threshold models. Journal of Cleaner Production, 286, 124992.

Shao, W., Yang, K., & Bai, X. (2021). Impact of financial subsidies on the R&D intensity of new energy vehicles: A case study of 88 listed enterprises in China. Energy Strategy Reviews, 33, 100580.

Reviewer #2: make correction according to the review report comments, The article's title is noteworthy, the presentation is strong, and it may offer a number of scientific arguments. The following recommendations will help writers contribute more to technological research and probability.

6. PLOS authors have the option to publish the peer review history of their article (what does this mean?). If published, this will include your full peer review and any attached files.

Reviewer #1: No

Reviewer #2: No

---

## [Author Response · Author response to Decision Letter 0]

14 Feb 2023

Dear reviewers and editor,

First of all, thank you for your valuable comments, in response to your comments this article has been revised as follows.

Reviewer #1:

1.The current introduction seems to provide too much background information regarding China NEV industry while the motivation and contribution are not made clear. I suggest the following: 

Shorten the background information, brief review of the relation between government subsidies and firm innovation in general (i.e., existing evidence of non-linear relationship, crowding-out effect in this relationship), 

State the research subject of the paper in the first/ second paragraph of the introduction which is different with the existing studies as it first provides the evidence of inverted U-shape relationship on NEV firm innovation in China. 

State the reason why it is important to explore how the effect of government subsidies varies by the role of firms in value chain, the firm type, and the firm age. This would help highlight the main motivation and contribution of the paper in introduction section.

Response: The introduction has reduced the description of the background information, highlighted the theme of the study, explained the proposed inverted "U" relationship, added a description of the heterogeneity analysis section and the motivation of the study, and emphasized the contribution of the study.

2.In literature review, the paper provides appropriate hypothesis developments. However, regarding the hypothesis 1, I suggest citing more relevant and recent literature as mentioned above. Please check the duplicate sentences (which happen quite frequently) and redraft the literature review section.

Response: The literature review section was extensively revised to provide more arguments for the hypothesis, more recent references were cited, and sentence repetition was thoroughly revised.

3.Regarding variable selection, I perceive that there are many wrong descriptions of the variables used in the model. For example, “Patent¬¬it is the explanatory variable”??? More importantly, the authors did not provide the reasons why they only use Patent as a proxy for firm innovation. Since the major existing study documents that firm innovation is an input-output process which could be proxied by R&D spending and the number of the patents. (Li et al., 2021). On this matter, I would prefer to see more detailed explanation on the selection of the dependent variable. Turning to the selection of control variables, the paper should cite more relevant papers which document the correlations between there control variables with firm innovation. There is a lack of description of 𝛾¬¬¬¬t and μ_i variables in the model (Are they denotations of year fixed effect and firm fixed effect, respectively?)

Response: Each of the selected variables is described in detail, and more references are cited as the basis for the selection of variables, and the model explanation is re-stated.

4.Methodology: Existing studies have extensively recognized that there are potential endogeneity issues in the regression of firm innovation on subsidies. That means the subsidy is the possible endogenous explanatory variable in the regression (Clausen, 2009 and Li et al., 2021). To address the potential endogeneity issues, I suggest conducting the Difference-in-Difference (DiD) test first, then the authors can run the Difference GMM and System GMM to estimate the model instead of using the static regression as the main methodology. As I perceive that the authors have employed GMM models in robustness checking, I suggest moving this empirical result to the main analyses and citing more supporting literature on the potential endogeneity issue of subsidy variable (the current paper does not cite the relevant studies regarding the issue of endogenous explanatory variable)

Response: For possible endogeneity issues, this paper reinserts two robustness tests, instrumental variation and replacement of explanatory variables, in the robustness test section to mitigate endogeneity. The dynamic panel estimation is moved to the underlying regression. For your proposed DID test, since this paper focuses on the non-linear relationship, it may not be quite applicable to the DID method for the time being, and we did not find any arguments about the DID estimation method for the non-linear relationship, so I hope you will make further pointers. In addition, regional heterogeneity was included in the heterogeneity analysis section, taking into account the regional variability.

5.Analyses of empirical results: The authors do not link the results of the paper to relevant findings in the extant literature. As such, the paper’s results remain isolated and then its contribution is limited.

Response: This revision enhances the interpretation of the empirical results and provides more literature support for the validation of the study results.

6.Policy implications: The current content of policy implication section is unlinked to the paper’s findings and somewhat irrelevant to the study. This way, the paper’s results do not provide insights for policy makers. I would prefer to see each policy implication is directly linked to the finding of the paper.

Response: Strengthened the correlation between the policy implications section and the study results by making a one-to-one correspondence between the study results and the policy implications.

Reviewer #2

1.The author must clarify and elaborate more why studying different dimensions of work engagement adds more theoretical contribution to prior study that used aggregate measure of work engagement.

Response: The innovations and marginal contributions of the study are highlighted in the introduction and literature review section, and more references have been added.

2.Some more recent reference to support the benefits of supervisor support will need to be added.

Response: More recent references have been added to highlight the marginal contribution of the study.

3. The key assumptions regarding normality need to be explained. It is important to summarize key findings of their study in the abstract. This is important to help readers understand where the paper is going.

Response: The abstract has been reworked to increase the organization of the abstract.

4. The policy implications should be strengthened.

Response: Adjustments were made to the policy implications and to correspond to the findings of the study.

5. Carefully proofread the article.

Response: Thoroughly revised the article for language errors that appeared in the article.

6. Numbered all equation in the text sequentially

Response: The formulas that appear in the article are numbered in order.

7. The discussion of the results needs more elaboration. Particularly, more solid discussion how the results advance knowledge in prior research is required

Response: The interpretation of the empirical results is added and correlated with the relevant literature.

8. Limitations and Recommendation for Future Research need more elaboration. Actually, limitations and recommendation for future research should be discussed in separated paragraphs.

Response: The conclusion section is reworked. Policy recommendations are re-stated.

9. There is no sampling method, how the data was collected was not being revealed. The study also used a single source data, which lead to common method variance, and author do nothing about that a lot more errors in the methodological section.

Response: The sample selection methods, data collection, and empirical methods are explained in more detail, and more references to the literature are provided to support the selection of variables.

10. Conclusion should add some latest references to prove the results

Response: More recent literature has been added to verify the reliability of the findings and to correlate with the previous literature.

Thank you again for your valuable suggestions for this revision.

---

## [Decision Letter · Decision Letter 1]

13 Mar 2023

PONE-D-22-30783R1Government subsidies and innovation in new energy vehicle companies: Empirical study of new energy vehicle listed companies based on Shanghai and Shenzhen A-sharesPLOS ONE

Dear Dr. tian,

Thank you for submitting your manuscript to PLOS ONE. After careful consideration, we feel that it has merit but does not fully meet PLOS ONE’s publication criteria as it currently stands. Therefore, we invite you to submit a revised version of the manuscript that addresses the points raised during the review process.

We look forward to receiving your revised manuscript.

Kind regards,

Hung Do

Academic Editor

PLOS ONE

Journal Requirements:

Additional Editor Comments:

The reviewers are generally happy with the revision. However, there are few minor points that they suggest the authors to address to polish the paper.

Reviewers' comments:

Reviewer's Responses to Questions

**Comments to the Author**

1. If the authors have adequately addressed your comments raised in a previous round of review and you feel that this manuscript is now acceptable for publication, you may indicate that here to bypass the “Comments to the Author” section, enter your conflict of interest statement in the “Confidential to Editor” section, and submit your "Accept" recommendation.

Reviewer #1: All comments have been addressed

Reviewer #2: All comments have been addressed

2. Is the manuscript technically sound, and do the data support the conclusions?

Reviewer #1: (No Response)

Reviewer #2: Yes

3. Has the statistical analysis been performed appropriately and rigorously? 

Reviewer #1: (No Response)

Reviewer #2: Yes

4. Have the authors made all data underlying the findings in their manuscript fully available?

Reviewer #1: (No Response)

Reviewer #2: Yes

5. Is the manuscript presented in an intelligible fashion and written in standard English?

Reviewer #1: (No Response)

Reviewer #2: No

6. Review Comments to the Author

Reviewer #1: Review report on the manuscript PONE-D-22-30783R1

In the revised manuscript, the authors have seriously addressed the suggested comments.

Thank you for your revision.

I just have several minor notes on the writing of the paper as follows:

1. You should present the findings of the paper in present tense, not in past tense as in the current revised version.

2. If possible, you should use professional editing and proof reading to enhance the writing of the paper. Too many writing errors need to be corrected. For example, “First, government subsidies have incentive effect on enterprise innovation[8] , second, government subsidies inhibit enterprise innovation[9], and third, there is no significant effect[10], most of these literatures only consider the linear relationship between government subsidies and enterprise innovation, and less literature has more in-depth research on nonlinear mechanism.” or “The possible reason is that because SOEs will have the underwriting guarantee from the government, they may form a dependence effect on innovation incentives due to this underwriting guarantee and will not be eliminated from the market due to the decline of innovation ability, while non-SOEs must continuously innovate their products to adapt to the severe market competition due to their self-sustaining characteristics, and the characteristics of market economy determine the inevitability of innovation incentives for non-SOEs.” Such the lengthy sentences need to be redrafted. Not limited in the given examples, please check the whole paper for the same errors.

3. Please note that the selling point of your paper is that it first provides the evidence of inverted U-shape relationship between government subsidies and firm innovation in NEV industry in China. Meanwhile, you didn’t mention this main contribution in the current introduction. Do not state that “less literature has more in-depth research on nonlinear mechanism” and thereby your paper fills this void in the literature. As the reviewed literature shows that there are abundant studies exploring the nonlinear relationship between subsidies and firm innovation in China (see the references in my last review report). Highlight the selling point of your study in the introduction.

4. My suggested DID model can be used together with your instrumental variable analysis to test the causal effect of subsidy on firm innovation with the aim of reducing the endogeneity concerns as a robustness check. However, as your study has conducted several robustness checks, I perceive that the paper provides sufficient evidence to state the consistency of the results. Even though no further robustness check is required in this paper, you can refer to the research of Black and Kim (2012) to see how several methods are combined to address the endogeneity concern in panel regression (For your interest only)

Reviewer #2: While they did a terrific job, the authors needed expert English proofreading. I contend that authors need to be required to have a processional proofread their works.

7. PLOS authors have the option to publish the peer review history of their article (what does this mean?). If published, this will include your full peer review and any attached files.

Reviewer #1: No

Reviewer #2: No

---

## [Author Response · Author response to Decision Letter 1]

20 Mar 2023

March 20, 2023

Dear reviewers and editor,

First of all, thank you for your valuable comments, in response to your comments this article has been revised as follows.

Reviewer #1:

1. You should present the findings of the paper in present tense, not in past tense as in the current revised version.

Response: The tense of the paper has been completely revised.

2. If possible, you should use professional editing and proof reading to enhance the writing of the paper. Too many writing errors need to be corrected. For example, “First, government subsidies have incentive effect on enterprise innovation[8] , second, government subsidies inhibit enterprise innovation[9], and third, there is no significant effect[10], most of these literatures only consider the linear relationship between government subsidies and enterprise innovation, and less literature has more in-depth research on nonlinear mechanism.” or “The possible reason is that because SOEs will have the underwriting guarantee from the government, they may form a dependence effect on innovation incentives due to this underwriting guarantee and will not be eliminated from the market due to the decline of innovation ability, while non-SOEs must continuously innovate their products to adapt to the severe market competition due to their self-sustaining characteristics, and the characteristics of market economy determine the inevitability of innovation incentives for non-SOEs.” Such the lengthy sentences need to be redrafted. Not limited in the given examples, please check the whole paper for the same errors.

Response: Writing errors in the paper have been proofread by professional editors

3. Please note that the selling point of your paper is that it first provides the evidence of inverted U-shape relationship between government subsidies and firm innovation in NEV industry in China. Meanwhile, you didn’t mention this main contribution in the current introduction. Do not state that “less literature has more in-depth research on nonlinear mechanism” and thereby your paper fills this void in the literature. As the reviewed literature shows that there are abundant studies exploring the nonlinear relationship between subsidies and firm innovation in China (see the references in my last review report). Highlight the selling point of your study in the introduction.

Response: This paper reorganizes the introduction to include "first provides evidence of an inverted U-shaped relationship between government subsidies and firm innovation in China's NEV industry" and removes "less literature has more in-depth research on nonlinear mechanism".

4. My suggested DID model can be used together with your instrumental variable analysis to test the causal effect of subsidy on firm innovation with the aim of reducing the endogeneity concerns as a robustness check. However, as your study has conducted several robustness checks, I perceive that the paper provides sufficient evidence to state the consistency of the results. Even though no further robustness check is required in this paper, you can refer to the research of Black and Kim (2012) to see how several methods are combined to address the endogeneity concern in panel regression (For your interest only).

Response: Thank you very much for your valuable suggestions. In this paper, instrumental variables are used to alleviate the endogeneity of the empirical results, and since this paper focuses on nonlinear mechanisms, the DID method is not used. For your proposed method of using a combination of DID and instrumental variables will be focused on in our future research

Reviewer #2

1. While they did a terrific job, the authors needed expert English proofreading. I contend that authors need to be required to have a processional proofread their works.

Response: Thank you for your valuable suggestions, this article has been proofread and corrected by professional editors.

Sincerely

Mingfu Tian

---

## [Decision Letter · Decision Letter 2]

6 Apr 2023

Government subsidies and innovation in new energy vehicle companies: An empirical study of new energy vehicle listed companies based on Shanghai and Shenzhen A-shares

PONE-D-22-30783R2

Dear Dr. tian,

We’re pleased to inform you that your manuscript has been judged scientifically suitable for publication and will be formally accepted for publication once it meets all outstanding technical requirements.

Kind regards,

Hung Do

Academic Editor

PLOS ONE

Additional Editor Comments (optional):

Reviewers' comments:

Reviewer's Responses to Questions

**Comments to the Author**

1. If the authors have adequately addressed your comments raised in a previous round of review and you feel that this manuscript is now acceptable for publication, you may indicate that here to bypass the “Comments to the Author” section, enter your conflict of interest statement in the “Confidential to Editor” section, and submit your "Accept" recommendation.

Reviewer #1: (No Response)

2. Is the manuscript technically sound, and do the data support the conclusions?

Reviewer #1: (No Response)

3. Has the statistical analysis been performed appropriately and rigorously? 

Reviewer #1: (No Response)

4. Have the authors made all data underlying the findings in their manuscript fully available?

Reviewer #1: (No Response)

5. Is the manuscript presented in an intelligible fashion and written in standard English?

Reviewer #1: (No Response)

6. Review Comments to the Author

Reviewer #1: Review report on the manuscript PONE-D-22-30783_R2

Thank you for your work to revise the paper. I perceive that the authors have addressed all the comments in my last review report. The writing of the paper has been enhanced a lot compared to the last version.

I have two minor suggestions to make your idea delivery more accurate, as follows:

1. In the abstract, the first finding is inaccurately stated. It should be corrected as: The paper finds that the government subsidies have significant non-linear effects on firm innovations in NEV industry with ample evidence of inversed U-shape in this relationship.

2. “However, from a nonlinear perspective, this study explores the nonlinear relationship between government subsidies and new energy vehicle enterprises based on the linear relationship”. This sentence makes readers confused what you mean. It should be expressed as: This study explores the nonlinear relationship between government subsidies and NEV innovation based on the linear regression model with quadratic term added as an independent variable in the model.

I would be happy for the paper to be published on Plos One after addressing the above-mentioned comments.

7. PLOS authors have the option to publish the peer review history of their article (what does this mean?). If published, this will include your full peer review and any attached files.

Reviewer #1: No

---

## [Editor Report · Acceptance letter]

11 Apr 2023

PONE-D-22-30783R2 

Government subsidies and innovation in new energy vehicle companies: An empirical study of new energy vehicle listed companies based on Shanghai and Shenzhen A-shares 

Dear Dr. Tian:

I'm pleased to inform you that your manuscript has been deemed suitable for publication in PLOS ONE. Congratulations! Your manuscript is now with our production department. 

Kind regards, 

on behalf of

Dr. Hung Do 

Academic Editor

PLOS ONE